# Burnout syndrome among frontline doctors of secondary and tertiary care hospitals of Bangladesh during COVID-19 pandemic

**Fahmida Rashid**[1]*, **Rabiul Alam Md. Erfan Uddin**[1], **H. M. Hamidullah Mehedi**[2], **Satyajit Dhar**[1], **Nur Hossain Bhuiyan**[1], **Md. Abdus Sattar**[1], **Shahanara Chowdhury**[1]

**1** Chittagong Medical College, Chattogram, Bangladesh, **2** Chattogram General Hospital, Chattogram, Bangladesh

* dr.fahmidaswati@gmail.com

## Abstract

### Introduction

During the COVID-19 pandemic, healthcare workers had a high workload and were exposed to multiple psychosocial stressors. However, a knowledge gap exists about the levels of burnout among Bangladeshi frontline doctors during this COVID-19 pandemic. The study investigated burnout syndrome (BOS) among frontline doctors in two public secondary and tertiary care hospitals in Chattogram, Bangladesh.

### Materials & methods

This cross-sectional study involved frontline doctors working at two hospitals treating COVID-19 and non-COVID patients from June to August 2020. A self-administered questionnaire that included Maslach Burnout Inventory for Human Services Survey (MBI-HSS) was used to capture demographic and workplace environment information. ANOVA and t-test were used to determine the statistical differences in the mean values of the three dimensions of MBI-HSS. Scores for three domains of burnout: emotional exhaustion (EE), depersonalization (DP), and personal accomplishment (PA) were calculated. Post-hoc analysis was done to identify the significant pair-wise differences when the ANOVA test result was significant. Multiple logistic regression was performed to determine the influence of factors associated with BOS.

### Results

A total of 185 frontline doctors were invited to participate by convenience sampling, and 168 responded. The response rate was 90.81%. The overall prevalence of BOS was 55.4% (93/168) (95% CI: 47.5% to 63.0%). Moderate to high levels of EE was found in 95.8% of the participants. High DP and reduced PA were observed in 98.2% and 97% of participants. Younger age (25–29 years), being female, and working as a medical officer were independently associated with high levels of burnout in all three domains. EE was significantly higher

**Data Availability Statement:** All relevant data are within the paper and its Supporting information files.

**Funding:** he author(s) received no specific funding for this work.

**Competing interests:** The authors have declared that no competing interests exist.

in females (P = 0.011). DP was significantly higher in medical officers, those at earlier job periods, and those working more than 8 hours per day.

## Conclusion

During the COVID-19 outbreak, BOS was common among Bangladeshi frontline doctors. Females, medical officers, and younger doctors tended to be more susceptible to BOS. Less BOS was experienced when working in the non-COVID ward than in the mixed ward.

## Introduction

The novel Coronavirus, SARS-CoV2, has led to a substantial death toll worldwide. Healthcare workers (HCWs) have been disproportionately affected, with World Health Organisation estimates from September 2021 placing the number of deaths among hospital staff between 80000 and 180000 [1]. Widespread underreporting means that the actual total is likely to exceed this [1].

COVID-19 has exposed HCWs to unusual physical and mental stresses [1]. Burnout syndrome (BOS) is a psychological condition caused by (chronic) workplace-related stresses specifically refers to such an occupational context [2]. According to Maslach and Jackson, it consists of three dimensions: emotional exhaustion (EE), depersonalization (DP), and feelings of reduced personal accomplishment (PA) [3]. Feelings of being "emotionally overextended and weary by one's work" are referred to as EE [3]. "An unfeeling and impersonal approach towards receivers of one's care or assistance" is described as DP [3]. Reduced PA refers to reduced 'feelings of competence and achievement in one's work with people' [3]. BOS refers specifically to phenomena in the occupational context and should not be applied to describe experiences in other areas of life [2]. A systemic review conducted in 2018, which included 176 studies, showed overall BOS prevalence in physicians was 48.7% [4], and a meta-analysis of BOS among residents found a prevalence of 51% [5].

Previous outbreaks of respiratory infections were associated with increased psychological morbidity among healthcare workers. In the severe acute respiratory syndrome (SARS) outbreak of 2003, emotional distress, depression, and anxiety occurred more frequently among frontline HCWs [6, 7]. Increased rate of psychological distress in HCWs was also reported during the H1N1 influenza pandemic (2009) and the Middle East respiratory syndrome (MERS-CoV) outbreak (2014) [1, 8, 9]. High levels of BOS have been reported among HCWs, especially in the emergency and Intensive Care Units (ICU) [10].

Factors contributing to an increased risk of BOS include excessive workload, high time pressure, high mortality rate, and lack of time to address patients' needs adequately [11]. These emotions might deteriorate during a pandemic because of the unknown nature of the sickness, dealing with numerous infected people, and the personal risk of contracting the virus. Burnout also negatively affects patients and coworkers since it increases the probability of making poor decisions, potential animosity toward patients, medical errors, challenging interpersonal interactions, and infections of HWCs' families [12, 13]. BOS is one of the primary factors affecting the quality of work and performance [14]. According to several systematic reviews, high levels of BOS in HCWs have been linked to less safe patient care [15, 16]. Specifically, burnout in HCWs has consistently shown a dose-response relationship with poorer patient safety outcomes [15]. It has also been associated with anxiety, depression, marital stress, early retirement, substance abuse, and suicide among HCWs [17].

The study was carried out six months following the onset of COVID-19 when there had been no reports of the pandemic's effects on BOS among HCWs. For health authorities to create interventions and policies to assist their employees better and prepare for future infectious disease outbreaks, they must have access to adequate data on BOS prevalence among HCWs.

The purpose of the study was to describe the BOS status of the frontline doctors in secondary and tertiary care institutions (mixed hospitals) in Bangladesh's south-eastern region.

## Materials& methods

### 1. Settings and participants

This cross-sectional study was carried out in Chittagong Medical College Hospital (CMCH) and Chattogram General Hospital (CGH), Chattogram, Bangladesh, from June to August 2020. CMCH is a 1313-bedded, tertiary-level teaching hospital that repurposed 200 beds for COVID-19 patients, including ICU, while keeping other services functional. CGH is a 250-bedded district general hospital repurposed with 140 beds, including ICU, for COVID-19 patients. The doctors in these hospitals provided services exclusively to COVID-19 patients, non-COVID-19 patients, or a mixture of COVID-19 and non-COVID-19 patients (termed 'mixed' in this study) following the principles of isolation and quarantine.

The researchers approached the prospective participants (convenience sampling), and those who had given informed written consent were included, and unwilling doctors were excluded from the study. The completed questionnaires were collected, verified, sorted, and analyzed.

### 2. Data collection tool

A self-administered questionnaire with two sections was used to collect the data. The first section included questions regarding socio-demographics and workplace details. The second section evaluated burnout using the Maslach Burnout Inventory for Human Services survey (MBI-HSS). The MBI-HSS consisted of 22 items and was divided into three domains: emotional exhaustion (EE, nine items), depersonalization (DP, five items), and personal accomplishment (PA, eight items) [3].

The items were measured with a 5-point Likert scale, from 1-never, 2-a few times a year, 3-a few times a month, 4-a few times a week, and 5-everyday. The numeric responses for each item were added to give the total score for each domain. Based on the scores, participants were categorized as having low ($\leq$18 points), moderate (19–26 points), high ($\geq$27 points) levels of EE; low ($\leq$5 points), moderate(6–9 points), high($\geq$10 points) levels of DP, and low ($\geq$ 40 points), moderate (39–34) and high($\leq$33) levels of reduced PA.

Although in Bangladesh, 25-item Shimul Burnout Inventory (SBI) (Bengali version) was used to measure job-related BOS administered in heterogeneous occupational categories [18]. As the medium of instruction for the graduation of frontline doctors was English in Bangladesh, the original MBI- HSS (english version) was used to collect the data.

The *Reliability Statistics* (reliability Alpha coefficient) for all items of MBI-HSS were included in the analysis. The Cronbach's Alpha value is 0.715 for the total scale. And the values for the EE, PA, and DP were 0.616, 0.602, and 0.145, respectively.

### 3. Ethical considerations

The research received approval from the ethical review committee of Chittagong Medical College, Bangladesh. All participants signed an informed consent form, and confidentiality was maintained.

## 4. Data analysis

Data were analyzed using SPSS statistical software version 22. To ascertain the statistical differences between the mean values of the three dimensions of the MBI-HSS, we employed the ANOVA and t-tests. When the ANOVA test result was significant, post-hoc analysis was carried out to pinpoint the important pair-wise differences. Multiple logistic regression analysis was conducted to ascertain the impact of parameters related to BOS. A p-value of $<0.05$ was considered significant.

## 5. Assessment of Burnout syndrome in the participants

Participants with high scores for EE ($\geq 27$), DP ($\geq 10$), and reduced PA ($\leq 33$) were designated as having a high degree of burnout. Factors associated with BOS were also determined.

# Results

## 1. Demographic and service-related characteristics of enrolled doctors

A total of 185 doctors were invited to participate in the study; 168 frontline doctors responded. The response rate was 90.81%. The overall prevalence of high BOS was 55.4% (93/168) (95% CI: 47.5% to 63.0%). Table 1 describes the demographic and service-related data of the respondents. The male-female ratio of the respondents was 106: 62 (1: 0.6), where about two-thirds were male and rest were female. About one-third of the respondents were 25–29 years, and the rest were >30 years old. More than three-quarters were government employees, and the remaining were private doctors from different specialties. About two-thirds were post-graduate students, and one-third were medical officers. Three-fifth of them was in their first to the

**Table 1. Demographic and service-related characteristics of the participants (n = 168).**

| Variables | | Frequency (percentage) |
|---|---|---|
| Age, years | 25–29 | 56 (33.3) |
| | 30–34 | 81 (48.2) |
| | >35 | 31 (18.5) |
| Sex | Male | 106 (63.1) |
| | Female | 62 (36.9) |
| Type of employment | Government | 131 (78.0) |
| | Private | 37 (22.0) |
| Course/Job designation | Post-graduate residents | 110 (65.5) |
| | Medical officer | 58 (34.5) |
| Academic year/job duration | $1^{st}$-$3^{rd}$year | 104 (61.9) |
| | $4^{th}$-$5^{th}$ Year | 64 (38.1) |
| Location of work | Obstetrics & Gynecology | 35 (20.8) |
| | Medicine | 61 (36.4) |
| | ICU | 35 (20.8) |
| | Surgery& emergency | 37 (22.0) |
| Duty place of doctors: | COVID | 70 (41.7) |
| | Non-COVID | 14 (8.3) |
| | Mixed | 84 (50.0) |
| Patient turnover/day: | <30/24 Hours | 103 (61.3) |
| | >30/24 Hours | 65 (38.7) |
| Duration of duty hour/day: | <8 Hours/Day | 77 (45.8) |
| | 8–12 Hours/Day | 91 (54.2) |

**Table 2. MBI subscale scores among the participants (n = 168).**

| MBI subscale | Mean±SD | Range |
|---|---|---|
| EE subscale | 27.7±5.5 | 13–45 |
| PA subscale | 21.7±4.9 | 8–36 |
| DP subscale | 16.7±2.9 | 9–24 |
| Total score | 66.21±13.51 | |

third year of academic study or employment, and the remainders were in later job/academic year periods (4th to 5th year). More than one-third worked in the medicine department. Two-fifth provided care for COVID-19 patients, and almost half were in the 'mixed' duty place.

## 2. MBI subscale scores among the participants

The mean MBI score was 66.2±13.5. The mean and standard deviation of the MBI score is represented in Table 2. Mean scores for EE, DP, and PA were 27.7±5.5, 16.7±2.9, and 21.7±4.9, respectively (Table 2).

## 3. Frequency of Burnout among participants

The EE, DP, and PA levels were grouped into mild, moderate, and high based on the defined criteria. Among participants, 95.8% of respondents had moderate to a high level of EE, and almost all of them had a high level of DP (98.2%) and reduced PA (97%) (Table 3).

## 4. EE, DP, and reduced PA in participants

There was no significant difference between age groups across the three domains of burnout. Significantly higher EE was observed in females than in males (p-value 0.011) (Table 4). The medical officers suffered more DP than post-graduate students (p-value 0.011). More DP was also observed in more junior doctors (1st–3rd year vs. 4th–5th year) (p-value 0.029). Respondents who worked >8 hours/day suffered from high BOS in all three domains.

## 5. High burnout in all three domains related to demographic and job-related characteristics

Younger participants (25–29 years) were 6.45 times more likely to have high burnout in all three domains than the participants >35 years (AOR: 6.45, 95% CI: 1.95–21.43; p = 0.002). Females had significantly higher burnout in all three domains than males (p-value 0.014). Medical officers had higher BOS than the post-graduate students (AOR: 3.55, 95% CI: 1.55–9.50; P = 0.011). Participants in non-COVID wards experienced less BOS than in mixed wards (OR: 0.22; 95% CI: 0.05–0.91; P = 0.036) (Table 5). Physicians with mixed workplace duty were 3.97 times more likely to have high BOS in all three domains than those who had duty in

**Table 3. Frequency of Burnout among participants (n = 168).**

| Burnout domains | Low | Moderate | High |
|---|---|---|---|
| Emotional exhaustion | 7 (4.2) | 63 (37.5) | 98 (58.3) |
| Reduced personal accomplishment | 0 (0) | 5 (3.0) | 163 (97.0) |
| Depersonalisation | 0 (0) | 3 (1.8) | 165 (98.2) |

Data were expressed as frequency (percentage).

**Table 4. Participants' level of emotional exhaustion, depersonalisation, and reduced personal achievement by demographic and job-related characteristics (n = 168).**

| Variables | Emotional exhaustion | | Reduced personal achievement | | Depersonalisation | |
|---|---|---|---|---|---|---|
| | Mean±SD | P value | Mean±SD | P value | Mean±SD | P value |
| Age, years | 28.77±5.29 | | 21.23±4.65 | | 16.71±3.49 | |
| 25–29 | 21.84±5.95 | 0.066 | 21.38±4.71 | 0.085‡ | 16.85±2.69 | 0.699‡ |
| 30–34 | 25.87±4.65 | | 23.51±6.00 | | 16.23±2.53 | |
| >35 | | | | | | |
| Sex | | | | | | |
| Male | 26.95±5.36 | 0.011* | 22.16±5.36 | 0.141† | 16.75±2.81 | 0.791† |
| Female | 29.21±5.68 | | 20.93±4.23 | | 16.63±3.19 | |
| Type of employment: | | | | | | |
| Government | 27.77±5.66 | 0.949 | 21.93±4.96 | 0.318† | 16.76±2.34 | 0.650† |
| Private | 27.63±5.32 | | 21.00±5.03 | | 16.51±2.98 | |
| Course/Job Designation | | | | | | |
| Post-graduate residents | 27.20±5.68 | 0.061 | 21.44±4.96 | 0.302† | 16.29±2.81 | 0.011†* |
| Medical officer | 28.89±5.23 | | 22.28±5.05 | | 17.50±3.06 | |
| Academic year/job period | | | | | | |
| 1st– 3rd year | 28.40±5.08 | 0.067 | 21.84±4.85 | 0.693† | 17.09±3.02 | 0.029†* |
| 4th– 5th year | 26.78±6.02 | | 21.53±5.25 | | 16.08±2.72 | |
| Duty place of doctors: | | | | | | |
| Obs & Gynae | 29.94±4.75 | | 22.28±4.99 | | 17.25±3.26 | |
| Medicine | 25.96±5.77 | 0.004* | 20.65±4.61 | 0.065‡ | 15.87±2.93 | 0.033‡* |
| ICU | 27.71±5.13 | | 23.37±6.33 | | 17.83±3.20 | |
| Surgery& emergency | 28.81±5.59 | | 21.40±3.72 | | 16.83±3.20 | |
| Duty place of doctors: | | | | | | |
| COVID ward | 28.04±5.23 | | 22.63±5.72 | | 17.51±2.55 | |
| Non-COVID ward | 25.71±4.03 | 0.347 | 19.78±4.67 | 0.081‡ | 16.28±4.06 | 0.010‡* |
| Mixed ward | 27.91±6.03 | | 21.29±4.24 | | 16.11±2.92 | |
| Patient turnout/day | | | | | | |
| <30/24 hours | 27.89±5.58 | 0.754 | 22.18±5.58 | 0.135† | 17.08±5.36 | 0.036†* |
| >30/24 hours | 27.61±5.60 | | 27.62±5.60 | | 16.11±3.00 | |
| Duration of duty hours/day | | | | | | |
| <8 hours/day | 26.71±5.64 | 0.021 | 20.64±4.58 | 0.009† | 16.19±2.73 | 0.039†* |
| 8–12 hours/day | 28.69±5.39 | | 22.65±5.17 | | 17.14±3.07 | |

‡ P values were obtained from the ANOVA test;

†p values were obtained from the Independent sample t-test.

* Significant

the non-COVID unit only (AOR: 3.97, 95% CI: 1.06–14.85, p = 0.040). There were no significant differences regarding the type of employment, course/job designations, academic year, placement of duty, patient turnout, and duration of duty hours.

## Discussion

The COVID-19 pandemic has negatively affected healthcare workers' physical and psychological well-being [1]. The current study demonstrated high levels of BOS among frontline doctors in Bangladeshi hospitals caring for COVID and non-COVID patients during the pandemic. Increased EE, DP, and reduced PA were observed six months after the COVID-19 outbreak in

**Table 5. High Burnout in all three domains and demographic and job-related characteristics (n = 168).**

| | High burnout in all three domains | | COR (95% CI for COR) | AOR (95% CI for AOR) | P value |
|---|---|---|---|---|---|
| | No (n = 75) | Yes (n = 93) | | | |
| Age, years | | | | | |
| 25–29 | 16 (21.3) | 40 (43.0) | 4.54 (1.78–11.59) | 6.45 (1.95–21.43) | 0.002* |
| 30–34 | 39 (52.0) | 42 (45.2) | 1.96 (0.83–4.61) | 2.37 (0.87–6.47) | 0.093 |
| >35 | 20 (26.7) | 11 (11.8) | 1 | 1 | |
| Sex | | | | | |
| Male | 55 (73.3) | 51 (54.8) | 1 | 1 | |
| Female | 20 (26.7) | 42 (45.2) | 2.66 (1.18–4.36) | 1.39 (0.59–3.24) | 0.450 |
| Type of employment: | | | | | |
| Government | 57 (76.0) | 74 (79.6) | 1.23 (0.59–2.55) | 1.03 (0.19–4.01) | 0.889 |
| Private | 18 (24.0) | 19 (20.4) | 1 | 1 | |
| Course/Job Designation | | | | | |
| Postgraduate residents | 55 (73.3) | 55 (59.1) | 1 | 1 | |
| Medical officer | 20 (26.7) | 38 (40.9) | 1.90 (0.98–3.67) | 3.55 (1.55–9.50) | 0.011* |
| Academic year | | | | | |
| 1st–3rd year | 41 (54.7) | 63 (67.7) | 1.74 (0.93–3.27) | 1.19 (0.54–2.60) | 0.672 |
| 4th and 5th year | 34 (45.3) | 30 (32.3) | 1 | 1 | |
| Placement of duty | | | | | |
| Obs. &Gynae | 10 (13.3) | 25 (26.9) | 1.525 (0.57–4.09) | 1.05 (0.30–3.61) | 0.939 |
| Medicine | 33 (44.0) | 28 (30.1) | 0.56 (0.22–1.19) | 0.39 (0.15–1.02) | 0.055 |
| ICU | 18 (24.0) | 17 (18.3) | 0.58 (0.23–1.47) | 0.67 (0.18–2.45) | 0.667 |
| Surgery& emergency | 14 (18.7) | 23 (24.7) | 1 | 1 | |
| Unit of duty place | | | | | |
| Non-COVID | 10 (13.3) | 4 (4.3) | 1 | 1 | |
| Mixed | 33 (44.0) | 51 (54.8) | 3.86 (1.12–13.34) | 3.97 (1.06–14.85) | **0.040*** |
| COVID | 32 (42.7) | 38 (40.9) | 2.97 (0.85–10.38) | 2.27 (0.58–8.89) | 0.241 |
| Patient turnout/ Day | | | | | |
| <30/24 Hours | 48 (64.0) | 55 (59.1) | 1.23 (0.65–2.30) | 1.11 (0.39–3.33) | 0.709 |
| >30/24 Hours | 27 (36.0) | 38 (40.9) | 1 | 1 | |
| Duration of duty hour/day | | | | | |
| <8 Hours/Day | 40 (53.3) | 37 (39.8) | 1 | 1 | |
| 8–12 Hours/Day | 35 (46.7) | 56 (60.2) | 1.73 (0.94–3.19) | 1.14 (0.75–4.12) | 0.335 |

*P values were obtained from Chi-square tests.

COR: Crude odds ratio; AOR: Adjusted odds ratio; CI: Confidence interval; 1 = Reference category

the country. During the study period, the country witnessed the pandemic's first peak, causing a rise in hospital admission of COVID-19 cases, including severe diseases. Frontline doctors became fatigued due to the increased workload and discomfort of wearing PPE. These findings agree with other studies on COVID-19 [19–21].

Studies that predate the emergence of COVID-19 have shown that 'large-scale natural disasters and pandemics are associated with significant increases in mental health disorders among health care providers' [22]. That might be due to the unprecedented challenge the healthcare system was unprepared to face [23]. Worldwide disruption of the health system [24] during the COVID pandemic increased morbidity and mortality even in developed countries, adding to the psychological distress of the HCWs. Another psychological problem that doctors experienced during the COVID pandemic was workplace violence [25].

In the current study, the overall prevalence of high BOS was 55.4% which was similar to that of Portugal (53%) [26], higher than that in other nations during the COVID-19 pandemic, including Brazil (21%) [27], Wuhan (FL 13% Vs. UW 39%) [28], Australia (30%) [29], Italy (37%,25%,15.3%) [30], Spain (41%,15.2%,8.4%) [31], and Egypt (35.5%, 70.6%, 26.5%) [32]. The difference might be due to cultural backgrounds, social factors, and different health system organizations. The small sample size and under-resourced infrastructure might be an additional factor in the current study.

The socio-demographic and work-related variables revealed that high BOS in all three domains was significantly more prevalent in the younger age group (25–29 years). They were more than six times more likely to have BOS than participants >35 years. Additionally, younger doctors (1st-3rd year vs. 4th-5th year) or early study levels suffered greater DP. This result agrees with a study conducted in Egypt [32] and India [33]. Similar findings were observed in the different studies conducted before the pandemic [34–36]. Because of less professional experience, younger and earlier job holders were more prone to anxiety and stress. And they often act as the first contact while dealing with COVID-19 patients [37]. However, a study on Jordanian nurses contradicts our findings, which show increased BOS among nurses with more education [38].

In this study, females were significantly more likely to experience burnout across all three domains. Particularly, EE was significantly more common in women than in men. This conclusion is in line with findings from other research [39, 40]. Studies in physicians before the pandemic also found that females suffered from more EE than their counterparts [12, 41]. In another study, being female predicts more EE [42]. In Bangladesh, female healthcare workers often have to play a dual role with a double workload, combining professional committee members with their domestic role, including caring for other family members. The high prevalence of EE in females might be due to this additional family stress and child-care commitments [43] and a less supportive work environment [12, 41].

EE and DP were significantly higher in the physicians working in the medicine department, who had to deal with most COVID-related deaths. EE, DP, and reduced PA were higher in gynaecology and obstetrics departments; they had to deal with COVID and non-COVID patients, which might cause high burnout. Martini et al. found the highest burnout rate in obstetrics and gynecology (75%), compared to 63% for internal medicine, 40% for general surgery, and the lowest in family medicine (27%) [44]. This difference might be because the units dedicated to COVID-19 in the hospitals were served predominantly by medical officers and residents from the medicine department. However, limited frontline physicians in the obstetrics and gynaecology department had to deal with more patients, which made it impossible to spare a dedicated group to deal with COVID-19 patients.

In this study, the physicians who worked in the mixed workplace duty showed more burnout in all three domains than in the non-COVID unit. A low level of burnout in frontline COVID unit medical professionals was observed in several studies compared to non-COVID wards in different centers [21, 27, 45]. An Egyptian study found high EE and DP in HCWs working in the COVID unit, but the difference was not statistically significant [32].

Working more than eight hours a day was significantly correlated with all three domains of burnout in the current study. This finding agrees with other studies conducted during the COVID-19 pandemic [32] and pre-pandemic era in Yemen, 2009, and Lebanon, 2010 [36, 46].

Burnout affects the quality of life and the delivery of health care services [32], which jeopardizes physical, mental, emotional, and social wellbeing. The high level of EE is related to a low level of mental health [47, 48] and needs effective intervention to combat it. A pandemic situation impairs the ability of physicians to maintain an appropriate work-life balance to combat anxiety and stress.

Since the study has shown that high burnout is more prevalent in Bangladesh than in many other settings, we have identified a dire need for a strategy to prevent it. A stress reduction program, which includes a program to reduce everyday stress and a system to cope with stress, seems to avoid burnout effectively [49]. Other individual-level interventions to improve resilience and coping methods with effective tools like online cognitive behavioral therapy appear effective [50]. But during the COVID-19 pandemic, system-level issues like—optimizing work quality and quantity should be addressed in Bangladesh.

Options to achieve these include interventions that reduce work inefficiencies, such as non-physician administrative support [51], satisfaction with work-flow, relationship with peers, spare time and resources for CME, opportunity to affect decision making, and a trusted advisor would reduce burnout [52].

And it was found that the strategy that addresses system-level matters is more effective. Physician burnout during the COVID-19 pandemic might be reduced if personal-level intervention could be combined with system-level intervention, enhancing job quality and quantity [51].

## Limitations

The scarcity of data on pre-pandemic burnout in the study hospitals for comparison was a limitation. The smaller sample size, inclusion of doctors solely from public (government) hospitals, and no data obtained from private (non-govt.) hospitals were other limitations. The effect of workplace factors for BOS's' job demand-control model' (JDCM) was not used here. Finally, the study was conducted during the peak of COVID-19 transmission when BOS was anticipated to be high.

## Conclusion

Our study demonstrated high levels of burnout among frontline doctors of different specialities during the COVID-19 pandemic in Bangladesh. BOS in females and younger were significantly more prevalent than in their counterparts. Doctors who worked in the non-COVID ward seemed to have less burnout than those in the mixed unit. Therefore, it is crucial to understand the prevalence of BOS and the associated risk factors to better support doctors working during this and future pandemic. A workplace mental health strategy and policy are essential for a healthy working environment during a pandemic crisis. Doctors who lack mental and emotional stability harm their patients, families, workplaces, and healthcare systems. Healthcare organizations and government authorities should develop mitigating strategies to ensure the well-being of doctors and other healthcare staff.

## Supporting information

**S1 File.**
(XLSX)

**S2 File.**
(XLSX)

**S3 File.**
(SAV)

**S4 File.**
(SAV)

**S1 Table. Post-hoc analysis.**
(DOCX)

# Acknowledgments

The authors would like to thank all volunteers for actively cooperating in the study. The authors are grateful to Dr. Farid Uddin Ahmed from the Department of Community Medicine, Chittagong Medical College, for his help in data analysis.

# Author Contributions

**Conceptualization:** Fahmida Rashid, Md. Abdus Sattar.

**Data curation:** Fahmida Rashid, Rabiul Alam Md. Erfan Uddin, H. M. Hamidullah Mehedi, Satyajit Dhar, Nur Hossain Bhuiyan, Md. Abdus Sattar, Shahanara Chowdhury.

**Methodology:** Fahmida Rashid, Rabiul Alam Md. Erfan Uddin, H. M. Hamidullah Mehedi, Satyajit Dhar, Nur Hossain Bhuiyan, Md. Abdus Sattar, Shahanara Chowdhury.

**Project administration:** Fahmida Rashid.

**Supervision:** Fahmida Rashid, Rabiul Alam Md. Erfan Uddin.

**Writing – original draft:** Fahmida Rashid, Rabiul Alam Md. Erfan Uddin, H. M. Hamidullah Mehedi, Satyajit Dhar, Nur Hossain Bhuiyan.

**Writing – review & editing:** Fahmida Rashid, Satyajit Dhar, Nur Hossain Bhuiyan, Md. Abdus Sattar, Shahanara Chowdhury.

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
