## [Decision Letter · Decision Letter 0]

3 Aug 2021

PONE-D-21-11241

Burnout Syndrome Among Frontline Doctors of COVID Hospitals During COVID-19 Pandemic, Bangladesh

PLOS ONE

Dear Dr. Rashid,

Thank you for submitting your manuscript to PLOS ONE. After careful consideration, we feel that it has merit but does not fully meet PLOS ONE’s publication criteria as it currently stands. Therefore, we invite you to submit a revised version of the manuscript that addresses the points raised during the review process.

The reviewers have identified a number of aspects of your manuscript that require substantial revision in order to meet PLOS ONE's publication criteria. Please address each of their points carefully when preparing your revisions, paying particular attention to clarifying the sample size and incorporating additional references to provide adequate context to your study.

We look forward to receiving your revised manuscript.

Kind regards,

Jamie Males

Staff Editor

PLOS ONE

Journal Requirements:

**3.**  We note you have included a table to which you do not refer in the text of your manuscript. Please ensure that you refer to Table 5 in your text; if accepted, production will need this reference to link the reader to the Table.

Reviewers' comments:

Reviewer's Responses to Questions

**Comments to the Author**

1. Is the manuscript technically sound, and do the data support the conclusions?

Reviewer #1: Yes

Reviewer #2: Partly

2. Has the statistical analysis been performed appropriately and rigorously? 

Reviewer #1: Yes

Reviewer #2: Yes

3. Have the authors made all data underlying the findings in their manuscript fully available?

Reviewer #1: Yes

Reviewer #2: Yes

4. Is the manuscript presented in an intelligible fashion and written in standard English?

Reviewer #1: No

Reviewer #2: Yes

5. Review Comments to the Author

Reviewer #1: The work provides an important insight to the challenges health professionals are facing in Bangladesh at the moment.

I suggest the authors the following improvements.

1. The paper requires some editing, as it is not written in sound English and should not be accepted in its current form. The author(s) may get the manuscript revised by either a native English-speaker or someone who excels in English language for this purpose.

2. The introduction section is well-defined with adequate references.

3. The materials and methods section is described in detail. However, the results section has not been interpreted properly.

4. There are places, although not many, in the discussion section where relevant references should have been provided to strengthen the findings. Besides, a more comprehensive discussion on what needs be done for an improved outcome could enhance the value of the work.

Reviewer #2: Thank you for giving me the opportunity to review this manuscript.

The authors address a relevant and topical issue, such as the impact that work on the front line against covid has on health professionals.

However, the manuscript can clearly be improved. The international literature on burnout in health professionals is extensive, as is the literature referring specifically to burnout during the COVID pandemic. In order for this article to be publishable, I believe that the authors must still make an important effort to go deeper both in the background and in the discussion. To do this, they must substantially expand the international literature they handle and reference (in the current version only 14 references, of which 6 are more than ten years old).

Regarding the methodology and presentation of results, it is a longitudinal descriptive study and the results are presented in an adequate manner, although they could have been enriched by adding some additional variable (perceived stress, workload, quality of life ... .), which would allow a greater depth in the discussion and conclusions.

There are small errors about the sample size (in method it is said that there are 165 professionals and in results it is said that 168).

For all of them I believe that in its current version this manuscript does not meet the minimum requirements for publication, although I believe that the authors should continue to delve into this topic of such interest.

6. PLOS authors have the option to publish the peer review history of their article (what does this mean?). If published, this will include your full peer review and any attached files.

Reviewer #1: **Yes: **Shaharior Rahman Razu

Reviewer #2: No

---

## [Author Response · Author response to Decision Letter 0]

12 Sep 2021

Response to Staff Editor Comments: 

Response: We have formatted the revised manuscript and named the files according to the journal style according to the suggestion.

"Upon re-submitting your revised manuscript, please upload your study's minimal underlying data set as either Supporting Information files or to a stable, public repository and include the relevant URLs, DOIs, or accession numbers within your revised cover letter. For a list of acceptable repositories, please see http://journals.plos.org/plosone/s/data-availability#loc-recommended-repositories. Any potentially identifying patient information must be fully anonymized.

Response: Data Availability Statement: The analyzed data is all in the manuscript. Four files of analyzed data uploaded as supporting file.

3. We note you have included a table to which you do not refer in the text of your manuscript. Please ensure that you refer to Table 5 in your text; if accepted, production will need this reference to link the reader to the Table.

Response: According to the suggestion, we have referred the Table 5 in the text of the results section (Page 12, line 168). 

Responses to Reviewer #1 

1. The paper requires some editing, as it is not written in sound English and should not be accepted in its current form. The author(s) may get the manuscript revised by either a native English speaker or someone who excels in the English language for this purpose.

Response: Thanks for the comments. According to the suggestion, we have taken help from a colleague of the English Department of a Local University and a senior colleague experienced in scientific writing from our institute. With their use, we have revised the entire manuscript to avoid language and grammatical errors.

2. The introduction section is well-defined with adequate references.

Response: Thank you, we appreciate the comments. 

3. The materials and methods section is described in detail. However, the results section has not been interpreted properly.

Response: Thank you for your valuable comments. According to your suggestion, we revised the results section and interpreted the results elaborately.

4. There are places, although not many, in the discussion section where relevant references should have been provided to strengthen the findings. Besides, a more comprehensive discussion on what needs to be done for an improved outcome could enhance the value of the work.

Response: Thanks for the remark, and we agree with that. We included few additional references to strengthen the findings.

Responses to Reviewer #2

Reviewer #2: Thank you for allowing me to review this manuscript.

The authors address a relevant and topical issue, such as the impact working on the front line against covid has on health professionals.

However, the manuscript can be improved. The international literature on burnout in health professionals is extensive, as is the literature referring specifically to burnout during the COVID pandemic. For this article to be publishable, the authors must still make an important effort to go deeper both in the background and in the discussion. To do this, they must substantially expand the international literature they handle and reference (in the current version, only 14 references, of which six are more than ten years old).

The methodology and presentation of results are a longitudinal descriptive study, and the results are presented adequately. However, they could have been enriched by adding additional variables (perceived stress, workload, quality of life ... .), which would allow a greater depth in the discussion and conclusions.

There are small errors about the sample size (in the method, there are 165 professionals, and in results, it is said that 168).

For all of them, I believe that this manuscript does not meet the minimum requirements for publication in its current version. However, I think that the authors should continue to delve into this topic of such interest.

Response: 

According to your suggestion, we made an effort to review related recent international literature on burnout in health professionals. After reviewing those articles, the background and discussion sections were revised and edited accordingly. 

Regarding your suggestion of adding some additional variables, we will keep it in our mind for our future work. 

The sample size was 168, corrected in the methods section (Page 5, line 94). There are minor errors about the sample size (in the method, it is said that there are 165 professionals, and in results, it is said that 168).Thanks for marking it.

---

## [Decision Letter · Decision Letter 1]

10 Mar 2022

PONE-D-21-11241R1Burnout Syndrome Among Frontline Doctors of COVID Hospitals During COVID-19 Pandemic, BangladeshPLOS ONE

Dear Dr. Rashid,

Thank you for submitting your manuscript to PLOS ONE. After careful consideration, we feel that it has merit but does not fully meet PLOS ONE’s publication criteria as it currently stands. Therefore, we invite you to submit a revised version of the manuscript that addresses the points raised during the review process.

The reviewer are not satisfied yet with the language and grammar of manuscript. Therefore, it is requested to carefully check the language of manuscript or take help from expert proofreader. If you have already did proofreading then you may need native speaker expert or professional proofreader.

We look forward to receiving your revised manuscript.

Kind regards,

Muhammad Shahzad Aslam, Ph.D.,M.Phil., Pharm-D

Academic Editor

PLOS ONE

Additional Editor Comments (if provided):

It is recommended to improve the language of manuscript and go-to experts for proofreading and resubmit.

Reviewers' comments:

Reviewer's Responses to Questions

**Comments to the Author**

1. If the authors have adequately addressed your comments raised in a previous round of review and you feel that this manuscript is now acceptable for publication, you may indicate that here to bypass the “Comments to the Author” section, enter your conflict of interest statement in the “Confidential to Editor” section, and submit your "Accept" recommendation.

Reviewer #2: All comments have been addressed

2. Is the manuscript technically sound, and do the data support the conclusions?

Reviewer #2: Yes

3. Has the statistical analysis been performed appropriately and rigorously? 

Reviewer #2: Yes

4. Have the authors made all data underlying the findings in their manuscript fully available?

Reviewer #2: Yes

5. Is the manuscript presented in an intelligible fashion and written in standard English?

Reviewer #2: Yes

6. Review Comments to the Author

Reviewer #2: The authors have made a great effort to answer the questions raised in the previous review and the manuscript has improved substantially.

I recommend that authors cite or reference the article on the Bangladeshi version of the Maslach questionnaire (in addition to the original citation already included).

I also recommend reviewing the English, especially in the introduction

7. PLOS authors have the option to publish the peer review history of their article (what does this mean?). If published, this will include your full peer review and any attached files.

Reviewer #2: No

---

## [Author Response · Author response to Decision Letter 1]

18 Apr 2022

Dear

Muhammad Shahzad Aslam, Ph.D.,M.Phil., Pharm-D

Academic Editor

PLOS ONE and dear reviewer 

Thank you very much. 

The manuscript has been revised as per your valuable suggestion. 

Hope this might fulfil your expectation.

If you still need any clarification, feel free to communicate.

---

## [Decision Letter · Decision Letter 2]

17 Jun 2022

PONE-D-21-11241R2Burnout Syndrome Among Frontline Doctors of COVID Hospitals During COVID-19 Pandemic, BangladeshPLOS ONE

Dear Dr. Rashid,

Thank you for submitting your manuscript to PLOS ONE. After careful consideration, we feel that it has merit but does not fully meet PLOS ONE’s publication criteria as it currently stands. Therefore, we invite you to submit a revised version of the manuscript that addresses the points raised during the review process. Please submit your revised manuscript by August 1 2022. If you will need more time than this to complete your revisions, please reply to this message or contact the journal office at plosone@plos.org. Please include the following items when submitting your revised manuscript:A rebuttal letter that responds to each point raised by the academic editor and reviewer(s). You should upload this letter as a separate file labeled 'Response to Reviewers'.A marked-up copy of your manuscript that highlights changes made to the original version. You should upload this as a separate file labeled 'Revised Manuscript with Track Changes'.An unmarked version of your revised paper without tracked changes. You should upload this as a separate file labeled 'Manuscript'.

We look forward to receiving your revised manuscript.

Kind regards,

Muhammad Shahzad Aslam, Ph.D.,M.Phil., Pharm-D

Academic Editor

PLOS ONE

Reviewers' comments:

Reviewer's Responses to Questions

**Comments to the Author**

1. If the authors have adequately addressed your comments raised in a previous round of review and you feel that this manuscript is now acceptable for publication, you may indicate that here to bypass the “Comments to the Author” section, enter your conflict of interest statement in the “Confidential to Editor” section, and submit your "Accept" recommendation.

Reviewer #2: (No Response)

Reviewer #3: (No Response)

Reviewer #4: All comments have been addressed

2. Is the manuscript technically sound, and do the data support the conclusions?

Reviewer #2: Yes

Reviewer #3: Partly

Reviewer #4: Partly

3. Has the statistical analysis been performed appropriately and rigorously? 

Reviewer #2: Yes

Reviewer #3: Yes

Reviewer #4: Yes

4. Have the authors made all data underlying the findings in their manuscript fully available?

Reviewer #2: Yes

Reviewer #3: Yes

Reviewer #4: Yes

5. Is the manuscript presented in an intelligible fashion and written in standard English?

Reviewer #2: Yes

Reviewer #3: No

Reviewer #4: Yes

6. Review Comments to the Author

Reviewer #2: I recommend that authors cite or reference the article on the Bangladeshi version of the Maslach questionnaire (in addition to the original citation already included). If they have used an adaptation / translation made ad-hoc for this study, it is something that should be specified, including at least reliability data (total Cronbach's alpha and by dimensions)

Reviewer #3: Dear authors. I think this paper still needs major revision again. English language editing is preferred. My comments are as follows

Abstract:

- Methods: Add how you select your data randomly, convenience..

- Please removed this paragraph """scores for three domains....... and replace it with statistical analysis you used.

Keywords: add doctors

Introduction

- Please set all references behind the full stop dot.

- You mention burnout syndrome several times before you mention the abbreviation in line 57. Please make sure to be consistent with your words.

- Add one paragraph related to your demographic factors.

Methods

- Again how you collect your data???. Add convenience sample procedure.

- What is your exclusion and inclusion criteria of your participants?

- Please add your response rate

- In which language you used the study tool?

- did you use Ch-square and t-test????add them on data analysis.

results

- replace sex to gender in all.

- you repeated what is already in Table 1. Give us a holistic view about your sample.

- Table, the letter t is capitalized when you mention Table.

- I notice that both dimension of DP, PA no one falls in low level. This result is surprising.

- I'm a quantified in this scale, EE is the main core of BOS, How can you explain that 58.3% of them had severe EE and 98.2% had severe BOS????

- Table 4. Please remove star over p-value... star over p-value mean significant not the test of what you used.

- Add reference category when you writing your result.

- Table 6, Add star on significant level.

Discussion

I suggest to remove all numbers of i.e., 58.3..

- add other psychological problem that doctors are suffered such as workplace violence. Read this paper "Workplace Violence among Healthcare Providers during the COVID-19 Health Emergency: A Cross-Sectional Study

- You found that higher educational lever are more prone to BOS. Please add more explanation. This paper can help. Self-evaluation and professional status as predictors of burnout among nurses in Jordan

- You mention in limitation section that is only from public hospitals?????????????? did you mention private

- References double check and remove hyperlinks.

Good luck

Reviewer #4: (No Response)

7. PLOS authors have the option to publish the peer review history of their article (what does this mean?). If published, this will include your full peer review and any attached files.

Reviewer #2: No

Reviewer #3: No

Reviewer #4: No

---

## [Author Response · Author response to Decision Letter 2]

29 Jul 2022

Response to reviewers attached as separate file

---

## [Decision Letter · Decision Letter 3]

19 Aug 2022

PONE-D-21-11241R3Burnout Syndrome Among Frontline Doctors of Secondary and Tertiary Care Hospitals of Bangladesh During COVID-19 PandemicPLOS ONE

Dear Dr. Rashid,

Thank you for submitting your manuscript to PLOS ONE. After careful consideration, we feel that it has merit but does not fully meet PLOS ONE’s publication criteria as it currently stands. Therefore, we invite you to submit a revised version of the manuscript that addresses the points raised during the review process.

We look forward to receiving your revised manuscript.

Kind regards,

Muhammad Shahzad Aslam, Ph.D.,M.Phil., Pharm-D

Academic Editor

PLOS ONE

Journal Requirements:

Reviewers' comments:

Reviewer's Responses to Questions

**Comments to the Author**

1. If the authors have adequately addressed your comments raised in a previous round of review and you feel that this manuscript is now acceptable for publication, you may indicate that here to bypass the “Comments to the Author” section, enter your conflict of interest statement in the “Confidential to Editor” section, and submit your "Accept" recommendation.

Reviewer #2: All comments have been addressed

Reviewer #3: (No Response)

Reviewer #4: All comments have been addressed

2. Is the manuscript technically sound, and do the data support the conclusions?

Reviewer #2: Yes

Reviewer #3: Yes

Reviewer #4: Yes

3. Has the statistical analysis been performed appropriately and rigorously? 

Reviewer #2: Yes

Reviewer #3: Yes

Reviewer #4: Yes

4. Have the authors made all data underlying the findings in their manuscript fully available?

Reviewer #2: Yes

Reviewer #3: Yes

Reviewer #4: Yes

5. Is the manuscript presented in an intelligible fashion and written in standard English?

Reviewer #2: Yes

Reviewer #3: (No Response)

Reviewer #4: Yes

6. Review Comments to the Author

Reviewer #2: (No Response)

Reviewer #3: Dear authors, your paper is enhanced significantly. However, minor issues still need more clarification before accepting this paper. English language editing is recommended.

Abstract

- conclusion part has to be written again. It's not a repeat of what in your results. Give a broad image of burnout during such crises among doctors. The implications for medicine must also be addressed.

Introduction

- Again, please set all references before the full stop dot.

- Add one paragraph related to your demographic factors.

Methods

- How did you collect your data by whom, and at what time, did they sign a consent form.... These issues need to be clarified.

Post hoc test: Which one??? Please add. Tukey, Sheffie....

Results

- In Table 5, why do you mention 2p-values.. Some of the p-values are significant without marks.

- Again, please change the word (sex) to gender in the whole of your manuscript.

The discussion part is improved significantly

Please return to the style of PLOS ONE reference guidelines, Some papers you added DOI, some references you add a month of publication.

Please double-check all references.

Good luck.

Reviewer #4: Comments for authors

Dear editors, thank you give the chance to review this paper and my comment is the following.

Reviewer Comments for a research entitled: Burnout Syndrome among Frontline Doctors of COVID Hospitals during COVID-19 Pandemic, Bangladesh

1. In the method section from line number 85 says the study was conducted from June to August 2020? Is it data collection period or the research is done with this time bound, if it is for data collection period I think it is so long 3 months for cross-sectional study data collection.

2. In the result section what was your criteria to categorize the age of your study participants line number 144(socio-demographic characteristics)?

3. In your data collection tool the first component is about socio –demographic and workplace details. Do you incorporate workplace factors for burn out syndrome like demand –control model (job control and job demand) model to see the effect of workload and workers capability to accomplished it.

4. Line number 175, High Burnout in All Three Domains and Demographic and Job-related and in it is regression table the category of <<type employment="" of="">> were not have AOR why it cannot candidate for multivariable regression model and it is unusual even the change of odds ratio should be seen from bi variable to multivariable model, so it is better to see again it.

5. In the discussion section after starting the discussion section with some statements that indicate the problem and next discussed the main finding of the your study therefore line number 198 -205 is part of you background or statement of the problem and it is better avoid it .

The paper is very important to provide essential information regarding health care workers level of burn out syndrome during outbreak management to develop an intervention.</type>

7. PLOS authors have the option to publish the peer review history of their article (what does this mean?). If published, this will include your full peer review and any attached files.

Reviewer #2: No

Reviewer #3: No

Reviewer #4: No

---

## [Author Response · Author response to Decision Letter 3]

30 Sep 2022

Responses to Reviewer's Questions

Reviewer #2: (No Response)

Reviewer #3: Dear authors, your paper is enhanced significantly. However, minor issues still need more clarification before accepting this paper. English language editing is recommended.

Abstract

- conclusion part has to be written again. It's not a repeat of what in your results. Give a broad image of burnout during such crises among doctors. The implications for medicine must also be addressed.

Introduction

- Again, please set all references before the full stop dot.

- Add one paragraph related to your demographic factors.

Methods

- How did you collect your data by whom, and at what time, did they sign a consent form.... These issues need to be clarified.

Post hoc test: Which one??? Please add. Tukey, Sheffie....

Results

- In Table 5, why do you mention 2p-values.. Some of the p-values are significant without marks.

- Again, please change the word (sex) to gender in the whole of your manuscript.

The discussion part is improved significantly

Please return to the style of PLOS ONE reference guidelines, Some papers you added DOI, some references you add a month of publication.

Please double-check all references.

Good luck.

Response to Reviewer #3 Comment:

-Tried my best to improve the editing of the English language. 

In abstract:

-Conclusion part of abstract rewritten according to your suggestion. Implication for medicine addressed.

Introduction:

- All references are set before the full stop dot.

- One paragraph is added related to my demographic factors.

Methods:

- How data was collected, by whom, at what time, and whether they signed a consent form. - These issues be clarified in the methods and material section under the heading-‘settings and participants'.

-Type of Post hoc analysis is added to results and discussion(Tukey HSD-Mentioned in Line 185 & 188).

Results:

-In table 5, 2 p-values were mentioned---One is for univariate analysis and an unadjusted odds ratio, and the other for the multivariate analysis-adjusted odds ratio. The significant p-values are marked by the star a (*) mark. 

- The word 'sex' changed to gender in the whole manuscript.

-Double-checked references according to PloS one guideline and did necessary modifications.

Reviewer #4: Comments for authors

1. In the method section from line number 85 says the study was conducted from June to August 2020? Is it data collection period or the research is done with this time bound, if it is for data collection period I think it is so long 3 months for cross-sectional study data collection.

Answer: Thank you very much. Because of frontline doctors' lockdown and quarantine policy, it wasn't easy to research the doctors for data collection. It took three months to collect and compile data from two institutes.

2. In the result section what was your criteria to categorize the age of your study participants line number 144(socio-demographic characteristics)?

Answer: Categorization of the age of respondents into different groups based on their academic year(1st-3rd and 4th-5th year) is described in the result section.

3. In your data collection tool, the first component is about socio –demographic and workplace details. Do you incorporate workplace factors for burn out syndrome like demand –control model (job control and job demand) model to see the effect of workload and workers capability to accomplished it.

Answer: In our data collection tool, in the first component, only socio-demographic profiles were included, and workplace details were not incorporated. So, in the current manuscript, erroneously mentioned words related to workplace details are discarded.

4. Line number 175, High Burnout in All Three Domains and Demographic and Job-related and in it is regression table the category of <> were not have AOR why it cannot candidate for multivariable regression model and it is unusual even the change of odds ratio should be seen from bi variable to multivariable model, so it is better to see again it.

Answer: In table -5(High burnout in all three domains in demographic and demographic and Job-related characteristics) - only the variables with p<0.2 in bivariate analysis were included in the multivariable analysis.

The change of OR is rechecked according to your suggestion, and the results are the same.

5. In the discussion section after starting the discussion section with some statements that indicate the problem and next discussed the main finding of the your study therefore line number 198 -205 is part of you background or statement of the problem and it is better avoid it .

Answer: Correction done as per your suggestion.

The paper is very important to provide essential information regarding health care workers level of burn out syndrome during outbreak management to develop an intervention.

Answer:

Thank you.

1. Please ensure that you refer to Table 6 in your text as, if accepted, production will need this reference to link the reader to the Table.

Answer: Table 6 referred to text according to your suggestion.

---

## [Decision Letter · Decision Letter 4]

10 Oct 2022

PONE-D-21-11241R4Burnout Syndrome Among Frontline Doctors of Secondary and Tertiary Care Hospitals of Bangladesh During COVID-19 PandemicPLOS ONE

Dear,

Thank you for submitting your manuscript to PLOS ONE. After careful consideration, we feel that it has merit but does not fully meet PLOS ONE’s publication criteria as it currently stands. Therefore, we invite you to submit a revised version of the manuscript that addresses the points raised during the review process. Please submit your revised manuscript by 9th November 2022. If you will need more time than this to complete your revisions, please reply to this message or contact the journal office at plosone@plos.org. Please include the following items when submitting your revised manuscript:A rebuttal letter that responds to each point raised by the academic editor and reviewer(s). You should upload this letter as a separate file labeled 'Response to Reviewers'.A marked-up copy of your manuscript that highlights changes made to the original version. You should upload this as a separate file labeled 'Revised Manuscript with Track Changes'.An unmarked version of your revised paper without tracked changes. You should upload this as a separate file labeled 'Manuscript'.If applicable, we recommend that you deposit your laboratory protocols in protocols.io to enhance the reproducibility of your results. Protocols.io assigns your protocol its own identifier (DOI) so that it can be cited independently in the future. For instructions see: https://journals.plos.org/plosone/s/submission-guidelines#loc-laboratory-protocols. Additionally, PLOS ONE offers an option for publishing peer-reviewed Lab Protocol articles, which describe protocols hosted on protocols.io. Read more information on sharing protocols at https://plos.org/protocols?utm_medium=editorial-email&utm_source=authorletters&utm_campaign=protocols.

We look forward to receiving your revised manuscript.

Kind regards,

Muhammad Shahzad Aslam, Ph.D.,M.Phil., Pharm-D

Academic Editor

PLOS ONE

Journal Requirements:

Reviewers' comments:

Reviewer's Responses to Questions

**Comments to the Author**

1. If the authors have adequately addressed your comments raised in a previous round of review and you feel that this manuscript is now acceptable for publication, you may indicate that here to bypass the “Comments to the Author” section, enter your conflict of interest statement in the “Confidential to Editor” section, and submit your "Accept" recommendation.

Reviewer #3: All comments have been addressed

Reviewer #4: (No Response)

2. Is the manuscript technically sound, and do the data support the conclusions?

Reviewer #3: Yes

Reviewer #4: Yes

3. Has the statistical analysis been performed appropriately and rigorously? 

Reviewer #3: Yes

Reviewer #4: Yes

4. Have the authors made all data underlying the findings in their manuscript fully available?

Reviewer #3: Yes

Reviewer #4: Yes

5. Is the manuscript presented in an intelligible fashion and written in standard English?

Reviewer #3: Yes

Reviewer #4: No

6. Review Comments to the Author

Reviewer #3: Congratulations. All of my comments have been added. My best wishes to authors for their effort. Great work

Reviewer #4: Dear editors , Thank you giving this opportunity to review this research paper the author address almost my questions but have the following point if the author can consider it.

1. In the result section what was your criteria to categorize the age of your study participants line number 144(socio-demographic characteristics in the previous question respond rather the author respond about academic position classification of the study participants and he/she can reflect again.

2. In the previous review the authors reflects this Answer: In table -5(High burnout in all three domains in demographic and demographic and Job-related characteristics) - only the variables with p<0.2 in bivariate analysis were included in the multivariable analysis, and in this regard the author is correctly respond and it is acceptable. But still in line number 176 and 177(regression table needs re write correctly such as Factors associated with high burnout in all three Domains… and in this table there are variables such as “Types of employment”, <<patient out="" turn="">>, that are present in bi-variable analysis but cannot present or cannot appear( eliminated) in multivariable regression model, meaning that have no report of adjusted odds ratio(AOR) either it is significant or insignificant value and when this appear recoding of this variable may be needed, the other point in this table is that writing p- value for COR is not needed( it is better avoid it).</patient>

7. PLOS authors have the option to publish the peer review history of their article (what does this mean?). If published, this will include your full peer review and any attached files.

Reviewer #3: No

Reviewer #4: No

---

## [Author Response · Author response to Decision Letter 4]

1 Nov 2022

Journal Requirements:

Answer: Reviewed reference list. There are no retracted references at present.No change has been made in the reference list apart from the style of reference writing.

Reviewer's Responses to Questions

Comments to the Author

1. If the authors have adequately addressed your comments raised in a previous round of review and you feel that this manuscript is now acceptable for publication, you may indicate that here to bypass the "Comments to the Author" section, enter your conflict of interest statement in the "Confidential to Editor" section, and submit your "Accept" recommendation.

Reviewer #3: All comments have been addressed

Reviewer #4: (No Response)

2. Is the manuscript technically sound, and do the data support the conclusions?

Reviewer #3: Yes

Reviewer #4: Yes

3. Has the statistical analysis been performed appropriately and rigorously?

Reviewer #3: Yes

Reviewer #4: Yes

4. Have the authors made all data underlying the findings in their manuscript fully available?

Reviewer #3: Yes

Reviewer #4: Yes

5. Is the manuscript presented in an intelligible fashion and written in standard English?

Reviewer #3: Yes

Reviewer #4: No

Answer: According to your prior suggestion, the manuscript was checked and improvised by a native English-speaking person(as I mentioned earlier). And any typographical or grammatical errors corrected, reformed as much as possible.

6. Review Comments to the Author

Reviewer #3: Congratulations. All of my comments have been added. My best wishes to authors for their effort. Great work

Reviewer #4: Dear editors , Thank you giving this opportunity to review this research paper the author address almost my questions but have the following point if the author can consider it.

1. In the result section what was your criteria to categorize the age of your study participants line number 144(socio-demographic characteristics in the previous question respond rather the author respond about academic position classification of the study participants and he/she can reflect again.

Answer: Thank you.

Here in our country, a physician graduated at least at the age of 24( including the intern period), so the lower limit of age taken as 25, and all resident/frontline doctors in government tertiary care hospitals are younger because of a need of in-service training for completion of postgraduation and most of them are usually around 35 years. That is why age was categorized as such. Also, the case records form was structured based on the previous study on burnout among resident doctors(Reference: Alhaffar, B.A., Abbas, G. & Alhaffar, A.A. The prevalence of burnout syndrome among resident physicians in Syria. J Occup Med Toxicol 14, 31 (2019). https://doi.org/10.1186/s12995-019-0250-0). And after analysis, the younger age group was found to have a 6.45 times higher chance of burnout than the older age group.

2. In the previous review the authors reflects this Answer: In table -5(High burnout in all three domains in demographic and demographic and Job-related characteristics) - only the variables with p<0.2 in bivariate analysis were included in the multivariable analysis, and in this regard the author is correctly respond and it is acceptable. But still in line number 176 and 177(regression table needs re write correctly such as Factors associated with high burnout in all three Domains… and in this table there are variables such as "Types of employment", <>, that are present in bi-variable analysis but cannot present or cannot appear( eliminated) in multivariable regression model, meaning that have no report of adjusted odds ratio(AOR) either it is significant or insignificant value and when this appear recoding of this variable may be needed, the other point in this table is that writing p- value for COR is not needed( it is better avoid it).

Answer: 

The Type of employment's P-value was not statistically significant in univariate analysis. That is why multivariate analysis did not calculate the P value for those non-significant variables. However, I remodeled the table according to your suggestions. Please check.

7. PLOS authors have the option to publish the peer review history of their article (what does this mean?). If published, this will include your full peer review and any attached files.

Do you want your identity to be public for this peer review? For information about this choice, including consent withdrawal, please see our Privacy Policy.

Reviewer #3: No

Reviewer #4: No

---

## [Decision Letter · Decision Letter 5]

6 Nov 2022

Burnout Syndrome Among Frontline Doctors of Secondary and Tertiary Care Hospitals of Bangladesh During COVID-19 Pandemic

PONE-D-21-11241R5

Dear,

We’re pleased to inform you that your manuscript has been judged scientifically suitable for publication and will be formally accepted for publication once it meets all outstanding technical requirements.

Kind regards,

Muhammad Shahzad Aslam, Ph.D.,M.Phil., Pharm-D

Academic Editor

PLOS ONE

---

## [Editor Report · Acceptance letter]

10 Nov 2022

PONE-D-21-11241R5 

Burnout Syndrome Among Frontline Doctors of Secondary and Tertiary Care Hospitals of Bangladesh During COVID-19 Pandemic 

Dear Dr. Rashid:

I'm pleased to inform you that your manuscript has been deemed suitable for publication in PLOS ONE. Congratulations! Your manuscript is now with our production department. 

Kind regards, 

on behalf of

Dr. Muhammad Shahzad Aslam 

Academic Editor

PLOS ONE